# Structural Insight into Catalysis by the Flavin-Dependent NADH Oxidase (Pden_5119) of *Paracoccus denitrificans*

**DOI:** 10.3390/ijms24043732

**Published:** 2023-02-13

**Authors:** Martin Kryl, Vojtěch Sedláček, Igor Kučera

**Affiliations:** Department of Biochemistry, Faculty of Science, Masaryk University, Kotlářská 2, 61137 Brno, Czech Republic

**Keywords:** FMN, NADH, dioxygen reduction, *Paracoccus denitrificans*

## Abstract

The Pden_5119 protein oxidizes NADH with oxygen under mediation by the bound flavin mononucleotide (FMN) and may be involved in the maintenance of the cellular redox pool. In biochemical characterization, the curve of the pH-rate dependence was bell-shaped with pK_a1_ = 6.6 and pK_a2_ = 9.2 at 2 μM FMN while it contained only a descending limb pK_a_ of 9.7 at 50 μM FMN. The enzyme was found to undergo inactivation by reagents reactive with histidine, lysine, tyrosine, and arginine. In the first three cases, FMN exerted a protective effect against the inactivation. X-ray structural analysis coupled with site-directed mutagenesis identified three amino acid residues important to the catalysis. Structural and kinetic data suggest that His-117 plays a role in the binding and positioning of the isoalloxazine ring of FMN, Lys-82 fixes the nicotinamide ring of NADH to support the *proS*-hydride transfer, and Arg-116 with its positive charge promotes the reaction between dioxygen and reduced flavin.

## 1. Introduction

Flavoenzymes are known, among many other biological activities, to be involved in the management of intracellular oxidative stress. A prototypical example is the mammalian NAD(P)H:quinone acceptor oxidoreductase 1 (NQO1), a flavin adenine dinucleotide (FAD)-containing enzyme that catalyzes the nicotinamide nucleotide-dependent reduction of a variety of compounds such as quinones (including lipophilic membrane-bound ubiquinone and vitamin E) and superoxides [1]. Similar proteins also exist in bacteria. The bacterium *Paracoccus denitrificans* produces an enzyme originally named “ferric reductase B” (FerB) [2] which behaves as a functional counterpart to NQO1 with respect to its reactivity with quinones [3,4] and superoxides [4,5]. An FerB-deficient mutant strain exhibited enhanced cellular sensitivity to oxidative stress [4]. The crystal structure of FerB has revealed that it forms a homodimer containing one bound flavin mononucleotide (FMN) per monomer [6].

A large scale proteomic study focused on elucidating the response to an induced oxidative stress has pinpointed two proteins, the products of the genes *pden_5119* and *pden_3133*, that show a homology to FerB and are significantly upregulated in the presence of the superoxide generator paraquat [7]. The three paralogs considerably differ in their catalytic activity for oxygen reduction. The NADH-oxidizing activity of Pden_5119 supplemented by FMN depends on the oxygen concentration in a saturable manner (*k*_cat_ = 55.8 ± 0.7 s^–1^, *K*_M_(O_2_) = 43 ± 1 μM), which qualifies the enzyme as an NADH oxidase [8]. On the contrary, oxygen is only a very weak substrate for FerB (*k*_cat_ = 0.244 ± 0.006 s^−1^, *K*_M_ = 130 ± 10 μM) [5]. The rate of NADPH oxidation by Pden_3133 in the presence of FMN is even oxygen-independent, suggesting that the reduced flavin formed in the active site is oxidized by oxygen only after its release from the enzyme [9].

The findings so far indicate that the Pden_5119 protein shares functional properties with both flavin reductases, which use flavin as a dissociable cosubstrate, and flavoprotein oxidases, which contain flavin as a permanently bound cofactor. It follows a sequentially ordered mechanism where FMN binds before NADH and the binary complex enzyme-reduced flavin formed after the hydride transfer persists for a long enough time to be oxidized by molecular oxygen [8]. The position of Pden_5119 as a possible “missing link” between the two canonical groups of flavin-dependent enzymes justifies a more detailed investigation into its structural particularities.

Here, we present a study of enzyme catalysis by Pden_5119 using enzymological and structural biology methods. We show evidence supporting a crucial role of three specific amino-acid residues for FMN cofactor binding and its interaction with substrates.

## 2. Results

### 2.1. pH Dependence Analysis

The catalytic activity was firstly analyzed by investigating the effect of pH. The pH-rate profile at a low (subsaturating) concentration of FMN (2 µM) was found to obey a bell-shaped curve, with apparent pK_a_s of 6.63 ± 0.08 for the ascending limb and 9.21 ± 0.08 for the descending limb. In contrast, the rate at a near-saturation FMN concentration of 50 µM was no longer pH-dependent at an acidic pH, leaving a single ionization curve with only a descending limb, displaying an apparent pKa value of 9.72 ± 0.04 (Figure 1).

### 2.2. Inactivation by Amino Acid Modifying Reagents

Given that the pKa value of histidine’s imidazole side chain in proteins is reported to be 6.6 ± 1 [10], our data support but do not prove the idea that deprotonation of a histidyl residue enables the enzyme to acquire full cofactor-binding capacity and catalytic activity. A confirmation was sought in experiments in which diethyl pyrocarbonate was used as a modifying reagent [11]. The incubation of the enzyme with millimolar concentrations of this reagent resulted in a progressive loss of the activity (Figure 2). Taking into account the experimentally determined rate constant for diethyl pyrocarbonate hydrolysis under our reaction conditions (*k*′ = 0.0203 min^−1^; see Appendix A), an analysis of the obtained data yielded the inactivation rate constant, *k*, of 0.25 ± 0.02 mM^−1^·min^−1^. Treatment of the inactivated enzyme with hydroxylamine almost completely restored the activity (Figure 3), suggesting that a histidine or tyrosine residue(s) had been carbethoxylated by diethyl pyrocarbonate.

The Pden_5119 enzyme used in this work is a recombinant fusion protein that contains 15 histidine residues per monomer, including those in the C-terminal hexahistidine tag. Figure 4A shows a series of UV difference spectra taken during its incubation with a 245-fold molar excess of diethyl pyrocarbonate. These spectra agreed in shape and peak position with that obtained after the modification of authentic free histidine (Figure 4B). There was only a very small negative change in the difference spectrum around 280 nm, arguing against a significant modification of tyrosine residues. A quantitative analysis of the spectra allowed us to calculate that a total of nine histidine residues per monomer became modified at the end of incubation. By a comparison of the time courses of changes in the degree of modification and enzymatic activity (the inset in Figure 4A), it can be seen that the modification of 3 His residues was enough for complete inactivation.

We also examined if substrates would protect the enzyme from inactivation by diethyl pyrocarbonate. Figure 5 shows that the addition of 1.6 mM FMN, but not of 1.6 mM NADH, exerted a clear protective effect. This result indicates that the inactivation was due to the modification of a residue at or near the binding site for the FMN cofactor. A complementary effect could also be observed. As detailed in Figure 6, prior modification by diethyl pyrocarbonate significantly decreased the affinity for FMN, which is manifested by a tenfold increase in the dissociation constant compared to unmodified protein.

The alkaline limbs present in both pH-rate profiles in Figure 1 may be attributable to the ionization of a group that must be protonated for catalysis. However, the observed pK_a_s gave no clear clue as to the identity of the crucial residue involved. A cysteine SH-group could be excluded, because there is no cysteine amino acid in the protein sequence. Treating the enzyme with reagents which modify lysine, tyrosine, and arginine residues produced results briefly described below and detailed in Supporting Materials.

Upon exposure to 10 mM pyridoxal 5’-phosphate, a reversible modifier of lysine [12], the activity decreased to a constant value in about 5 min. The final activity changed from 70% to 40% of the control level when the pH value of the reaction mixture was raised over the range of 8 to 10 (Appendix A). FMN (2 mM), but not NADH, effectively protected the enzyme against inactivation by pyridoxal 5’-phosphate.

1-Acetylimidazole is known to acetylate tyrosyl residues of proteins [13]. Appendix A shows the effect of the modification on the enzyme when different concentrations of 1-acetylimidazole were used. Applying the pseudo-first-order kinetic model with the correction to reagent hydrolysis (*k*′ = 0.055 min^−1^; see Appendix A), we obtained a rate constant, *k*, of (7.5 ± 0.9)·10^−3^ mM^−1^·min^−1^. Similar to the previous cases, FMN (2 mM) strongly blocked the inactivation.

Phenylglyoxal, an arginine-targeted agent [14], also caused time-dependent inactivation following the pseudo-first-order kinetics (Appendix A). The rate constant *k* for the inactivation by phenylglyoxal was estimated to be (3.9 ± 0.2)·10^−3^ mM^−1^·min^−1^. The presence of 2 mM FMN during the modification with 40 mM phenylglyoxal not only did not decrease k, but even increased it by about 50%.

### 2.3. Crystal and Solution Structure

In order to understand the structural basis for Pden_5119 catalysis, single crystals of this enzyme were prepared and used to collect diffraction data (resolution range 48.75–3.1 Å). The co-ordinates and experimental structure factors were deposited to the PDB with the accession code 7QW4. Statistics for the data refinement are provided in Appendix A. The monomer takes a flavodoxin-like fold with a central β sheet made up of five parallel strands in the order 21345 which is sandwiched between helices 1 and 5 on one side and helices 2, 3, and 4 on the other side.

The oligomeric state in solution was analyzed by means of small-angle X-ray scattering (SAXS). Ab initio modeling using the SAXS data obtained at a protein concentration of 1.5 mg·mL^−1^ (73 μM in monomer) yielded an envelope that has space for the dimer (Figure 7, Appendix A). Therefore, we can conclude that the dimer represents the major species in the solution.

### 2.4. Search for Homology Proteins

A DALI structural homology search (http://ekhidna2.biocenter.helsinki.fi/dali/, accessed on 12 February 2023) performed with the Pden_5119 monomer identified a number of proteins with Z scores ≥10, most of which were annotated as reductases for flavin, chromate, quinone, or azo dyes. A phylogenetic analysis of the genomic sequences of these structurally aligned proteins by Clustal Omega (https://www.ebi.ac.uk/Tools/msa/clustalo/ accessed on 12 February 2023) placed Pden_5119 into a clade with the flavin reductases EmoB from *Mesorhizobium* sp. BNC1 (PDB ID: 4LTD; [15]), SsuE from *E. coli* (PDB ID: 4PTY; [16]), MsuE from *Pseudomonas putida* (PDB ID: 4C76), and an oxidoreductase from *Corynebacterium diphtheriae* (PDB ID: 3K1Y) (Appendix A).

A summary comparison of the structures is presented in Table 1 and Figure 8.

### 2.5. Active Site Model

Despite co-crystallization and soaking trials, we could not obtain crystals of Pden_5119 containing bound FMN. The FMN molecule position was therefore predicted by in silico docking analysis (Figure 9). The binding energy and the corresponding dissociation constant were predicted to be −8.11 kcal·mol^−1^ and 1.4 μM, which are comparable to experimental results (Figure 6).

As can be seen in Figure 9, the putative active site contains a tyrosine residue (Tyr-81) that is juxtaposed to the *re*-side of the FMN isoalloxazine ring. This tyrosine is highly conserved among Pden_5119 homologues. Our previous study with the homologous flavoprotein FerB has shown that its presence enhances FMN binding but has little effect on the rate of reaction [6]. Further inspection of the isoalloxazine environment implicated three amino acid residues, His-117, Lys-82, and Arg-116, as potentially involved in catalysis. The imidazole ring of His-117 is positioned to form a hydrogen bond with the N3 nitrogen of FMN similar to the case for homologous His-112 and His-116 residues in SsuE and EmoB. In SsuE, Lys-77, an equivalent of Lys-82, was reported to be hydrogen-bonded to the O4 carbonyl atom of the second molecule of FMN stacked over the first one. The positively charged Arg-116 has no counterpart in SsuE and EmoB which contain an alanine at its position.

### 2.6. Site-Directed Mutagenesis and Mutant Characterization

To elucidate the role of the three selected amino acid residues, they were changed to alanine by site-directed mutagenesis and the C-terminally hexahistidine-tagged mutant proteins were produced in *E. coli*. The UV circular dichroism spectra of the products (Appendix A) have a similar shape as that of the wild-type protein, suggesting that the mutations had not caused any significant distortion of the secondary structure. All protein variants were still able to bind FMN, as demonstrated by the quenching of intrinsic protein fluorescence. The dissociation constants (*K*_d_^FMN^) determined by fluorometric titration are shown in Table 2. The lowest affinity for FMN was observed for H117A with the *K*_d_^FMN^, 2.4-fold higher than that for the wild-type enzyme.

During the Pden_5119-catalyzed reaction of NADH with O_2_, the bound FMN cofactor undergoes a cyclic conversion consisting of two half-reactions: reduction by NADH and oxidation by O_2_. Both half-reactions were analyzed separately by monitoring the time course of the oxidation state of the enzyme–FMN complex in a stopped-flow rapid-reaction apparatus at varying NADH or O_2_ concentrations. The experimental data could be fitted to a two-step kinetic model involving a rapid equilibrium-binding step followed by a rate-limiting redox reaction (Appendix A). Values for the fitted parameters *K*_NADH_, *k*_red_, *K*_O2_, and *k*_ox_ are given in Table 2 together with *k*_cat_ values determined by monitoring the steady-state rate of NADH consumption at saturation with NADH, FMN, and O_2_. By comparing the data, it can be seen that all three mutations severely impair the catalytic activity (lower *k*_cat_) but differ in their effect on individual parts of the flavin cycle. The correspondence of *k*_red_ with *k*_cat_ for the K82A mutant indicates that FMN reduction becomes the rate-limiting step in the catalytic turnover. On the other hand, the oxidative half-reaction may be rate-limiting in mutants H117A and R116A for which *k*_cat_ approaches *k*_ox_.

### 2.7. Kinetic Isotope Effect in Hydride Transfer from NADH

Kinetic assays were also performed with the selectively deuterated NADH derivatives [4*R*-^2^H]NADH (*R*-NADD) and [4*S*-^2^H]NADH (*S*-NADD). As shown in Table 3, the wild-type enzyme oxidized NADH and *R*-NADD at approximately equal rates, whereas the reaction rate with *S*-NADD was only about one-fourth of that with NADH. Since the C–D bond is stronger than the C–H bond, this observation can be interpreted as evidence for the hydride transfer from the *pro-S* position of NADH. A similar kinetic isotope effect was found with mutants H117A and R116A, but not with K82A, which apparently does not discriminate between *R*-NADD and *S*-NADD. This would be consistent with a role of Lys-82 in controlling the stereochemistry of the hydride transfer.

### 2.8. Effect of Arg-to-Lys Substitution at Position 116

We also were interested to find out whether the guanidine group of Arg-116 must specifically be present for the activity. We therefore introduced a charge-conserving change, lysine for arginine, at the position 116. The resulting mutant protein was found to retain almost the entirety of the wild-type activities in both the steady-state and pre-steady-state kinetic modes (the last line in Table 2). Arg-to-Lys replacement also led to a marked reduction of the ability of phenylglyoxal to inactivate enzyme. At the 40 mM concentration of phenylglyoxal, the *k*_obs_ for the mutant and wild type were 0.075 and 0.15 min^−1^, respectively (Appendix A). From these results, it can be concluded that a positive charge at position 116 by itself promotes oxygen reduction by NADH and that Arg-116 is a target (or one of the targets) for phenylglyoxal-derived modification.

## 3. Discussion

As revealed by our bioinformatic analysis (Appendix A, Figure 8, Table 1), the closest homologous relatives of Pden_5119 are flavin reductases associated with mono-oxygenases involved in sulfur acquisition from alkanesulfonates and methanesulfinate, and in EDTA degradation (reviewed in [18,19]). The enzymes SsuE, MsuE, and EmoB generate reduced FMN which is transferred to the respective mono-oxygenase via a protein–protein complex formation. The genes encoding these enzymes are parts of well-organized operons, which is not the case for the *pden_5119* gene. Pden_5119 differs from the above-mentioned flavin reductases functionally in that it lacks a mono-oxygenase partner and directs electron flow to oxygen. It also exhibits less tendency to oligomerization, being dimeric in solution (Figure 7), whereas the others were reported to be tetramers in their apoprotein state [15,16].

Our combined techniques-based approach taken in this study has pointed out three amino acid residues playing essential roles in the catalysis of the oxidase reaction.

### 3.1. Histidine-117

Evidence supporting His-117 involvement includes the p*K*_a_ of 6.6 on the pH-rate profile (Figure 1), the sensitivity to modification with diethyl pyrocarbonate (Figure 2, Figure 3 and Figure 4), and the very low overall catalytic activity of the H117A mutant (Table 2). The fact that the p*K*_a_ value appears only at a sub-saturating concentration of FMN is indicative of a protonic dissociation of a His residue in the free enzyme. The imidazole ring of His-117 in our structural model (Figure 9) as well as of homologous His residues 112 and 116 in SsuE (PDB ID: 4PTZ) and EmoB (PDB ID: 4LTM) is located in the vicinity of the N3 nitrogen of the isoalloxazine ring of the bound FMN. Since the N3 atom of FMN has a p*K*_a_ of about 10.8 [20], it exists predominantly in its protonated state at neutral pH and can act as a donor in hydrogen bonding with the deprotonated N-atom of the imidazole ring. Further support for the binding between histidine and FMN was provided by the protection against diethyl pyrocarbonate inactivation observed for the enzyme–FMN complex (Figure 5 and Figure 6). NADH alone cannot offer such protection consistently with the ordered sequential kinetic model [8] according to which it can bind to the enzyme only after FMN binding.

As follows from the performed spectrofluorimetric titration measurements (Table 2), the mutation H117A reduces, but does not prevent, FMN binding to the enzyme (a 2.4-fold increase in *K*_d_^FMN^, from 3.8 to 9.1 μM). The sluggish reaction rate under activity assay conditions where the FMN concentration (50 μM) is well above *K*_d_^FMN^, therefore, must have another reason than merely a low extent of saturation of the binding site. It is conceivable that His-117 helps to orient the isoalloxazine ring of the bound FMN into a position optimal for redox reactions. Given the *k*_red_ and *k*_ox_ values of the H117A mutant (Table 2), the proper ring position appears to be more critical for the oxidative than for the reductive half of the flavin redox cycle.

### 3.2. Lysine-82

The initial hint on participation of a lysine residue was deduced from the results on the inactivation by pyridoxal 5’-phosphate (Appendix A). The incomplete inactivation due to attaining binding equilibrium, its deepening with increasing pH, and its substrate-protective effect were already described previously for several other enzymes with lysine in their active site [12,21,22,23,24]. A rather similar pH dependence was also shown for the Schiff base formation of pyridoxal 5’-phosphate and α-amino acids [25].

Further support for a lysine’s role has come from the mutagenesis of Lys-82 and subsequent pre-steady-state kinetic and primary deuterium isotope effect studies (Table 2 and Table 3). The K82A mutant is more severely affected in FMN reduction than in oxidation, thus suggesting the involvement of Lys-82 in the interaction between NADH and FMN. Unfortunately, no direct structural data are currently available for a catalytically active complex of enzyme with FMN and NADH. The crystal structure of ternary complex EmoB-FMN-NADH published by Nissen et al. [15] (PDB ID: 2VZJ) was later corrected and shown to contain the adenine moiety of NADH instead of its nicotinamide part stacked on top of the isoalloxazine ring of FMN (PDB ID: 4LTN). EmoB and SsuE are known to bind a second molecule of FMN in a position allowing for electron and hydrogen atom transfers between both flavins [15,16,26] (PDB ID: 4LTM, 4PTZ). As illustrated in Figure 10A–C, besides π-stacked interactions, a hydrogen bond with a conserved lysine residue is important to the stabilization of these ternary complexes. We assume that the same lysine also forms a hydrogen bond with the oxygen of the carboxamide group of NADH, thereby orienting the *si*-face of the nicotinamide ring toward the isoalloxazine of FMN (Figure 10D). Removal of this H bond by mutation is expected to permit the free rotation of the nicotinamide ring accompanied by a loss in stereospecificity, which was indeed confirmed experimentally (Table 3).

### 3.3. Arginine-116

Several studies have demonstrated that the enrichment of positive charges in proximity of the N5 locus of the flavin can promote dioxygen reduction by accelerating the rate-limiting formation of a flavin semiquinone–superoxide radical pair (reviewed in [27]). In the case of Pden_5119, such favorable conditions are apparently created by the positively charged guanidinium group of Arg-116, which is located above the solvent accessible side of the isoalloxazine ring and can be functionally substituted by the positive ammonium group of lysine. This positive charge is essential for oxidase activity, as judged by the low *k*_cat_ and *k*_ox_ values of the R116A mutant (Table 2). Under the conditions of covalent modification experiments (Appendix A), 1:1 adducts are reported to be formed by the reaction between any one of the two carbonyl groups of phenylglyoxal and an unprotonated amino group of guanidine [28]. The observed decrease in activity following treatment with phenylglyoxal can be accounted for by a changed electrostatic state and steric hindrance for substrates to access the active site. Arg-116 itself seems not to directly participate in flavin binding, since the R116A mutation has no effect on the *K*_d_^FMN^ (Table 2) and the enzyme–FMN complex still remains modifiable by phenylglyoxal.

## 4. Materials and Methods

### 4.1. Recombinant Native and Mutant Protein Production and Purification

The cloning, expression, and purification of the Pden_5119 protein and its mutant forms with a C-terminal six-histidine tag in *E. coli* BL21(DE3)pLysS cells (ThermoFisher, Waltham, MA, USA) were carried out as described in the paper by Sedláček et al. [8]. QuickChange II Site-Directed Mutagenesis Kit (Agilent, Santa Clara, CA, USA) was used for site-directed mutagenesis. The primers for site-directed mutagenesis are listed in Appendix A. The sequences of the new mutated 5119 constructs were verified by sequence analysis. Correct folding of all mutant enzymes was confirmed by circular dichroism spectroscopy.

### 4.2. Determination of Protein Concentration

Enzyme protein was determined using Pierce BCA Protein Assay Kit (ThermoFisher Scientific, Waltham, MA, USA) according to instruction manual. The subunit molar concentration was calculated by dividing protein mass concentration (g·L^−1^) by the molar mass of the subunit (20 596 g·mol^−1^). Protein eluted from an FPLC apparatus (Cytiva, Marlborough, MA, USA) was monitored at 280 nm.

### 4.3. Enzyme Activity Assay

Enzyme activity was measured at 340 nm and 30 °C in an UltroSpec 2000 spectrophotometer (Cytiva, Marlborough, MA, USA). The standard assay mixture contained, in 1 mL, 150 μM NADH, 50 μM FMN, and 0.1 M sodium phosphate buffer (pH 7.3). Initial rates were calculated from the slope of the absorbance decrease at 340 nm by using a molar absorption coefficient of 6220 M^−1^·cm^−1^.

The pH effect was investigated in 40 mM Britton–Robinson buffer (a mixture of 40 mM boric acid, 40 mM phosphoric acid, and 40 mM acetic acid, adjusted to the desired pH with NaOH) containing 150 μM NADH and 2 μM or 50 μM FMN. The pH-rate dependence data were fitted to the single-ionization model
log *v* = (log *v*)_max_ − log (1 + 10^pH − p*K*a^)
or to the double-ionization model
log *v* = (log *v*)_max_ − log (1 + 10^p*K*a1 − pH^ + 10^pH − p*K*a2^)
where *K*_a_ represents the dissociation constants [29]. Non-linear fits were performed and standard errors of p*K*_a_s were calculated using Microsoft Excel 2016 as described in [30].

### 4.4. Chemical Modification

The involvement of histidine, lysine, tyrosine, and arginine residues in the catalytic action of the enzyme was examined by testing the effect of amino-acid-modifying reagents diethyl pyrocarbonate, pyridoxal 5’-phosphate, 1-acetylimidazole, and phenylglyoxal (Merck, Rahway, NJ, USA), respectively. Stock solution of diethyl pyrocarbonate was prepared by diluting in cold absolute ethanol and quantifying the actual concentration by reaction with 10 mM imidazole (Serva, Heidelberg, Germany) in 0.1 M sodium phosphate, pH 7.3, to form N-carbethoxyimidazole, with a molar absorption coefficient of 3000 M^- 1^·cm^−1^ at 230 nm [31]. The other reagents were dissolved in phosphate buffer. The reactions with enzyme were carried out at 30 °C or at 0 °C in 0.1 M sodium phosphate, pH 7.3. At various times, aliquots were removed and assayed for residual activity. When working with diethyl pyrocarbonate, the inactivation reaction in withdrawn samples was rapidly quenched by twofold dilution with 20 mM imidazole before activity assaying.

To determine the apparent reaction rate constant (*k*_app_) values for enzyme inactivation, a pseudo-first-order kinetics model was used in the form
ln(*A*(t)/A(0)) = − *k*_app_·*t*
where *k*_app_ is the apparent rate constant, and *A*(0) and *A*(t) are the activities at the beginning and after time *t* of the reaction. For a one-step inactivation mechanism, the apparent rate constant should increase linearly with the initial concentration of inactivator [I]_0_:*k*_app_ = *k* · [I]_0_

The slope of the resulting line provides the second-order rate constant (*k*) for inactivation [32].

Diethyl pyrocarbonate and 1-acetylimidazole hydrolyze in aqueous solutions. The first-order rate constant *k*′ for hydrolysis was determined by assaying time courses of changes in concentration in reaction buffer without enzyme. The progress of hydrolysis of diethyl pyrocarbonate was monitored by sampling into imidazole quench solution and measuring the absorbance at 230 nm as described above. The disappearance of 1-acetylimidazole was followed continuously at 243 nm. Taking into account the decomposition of the reagent, the rate equation for loss of enzyme activity is to be modified to [33]:ln(*A*(t)/A(0)) = ln*A*(0) − *k*_app_· (1 − e^−*k*′·*t*^)/*k*′

Each modification experiment was repeated several times with different concentrations of modifying agent and sampling intervals and the data from a typical experiment are presented.

### 4.5. Rapid Kinetics Measurements

Stopped-flow spectrophotometry was performed on an SFM-3000 stopped-flow system (Bio-Logic, Claix, France) equipped with a 0.8 × 0.8-mm cuvette (FC-0.8). The temperature was kept constant at 10 °C. Six to nine measurements were averaged for each sample. In one run, 75 μL of a protein solution with FMN was mixed with 75 μL of NADH under anaerobic conditions during the reduction phase or 75 μL of a protein solution with a chemically reduced FMN with 75 μL of the solution with different concentrations of oxygen during oxidation phase. For all experiments, 0.1 M sodium phosphate buffer (pH 7.3) was used as a standard reaction environment. Anaerobic buffer solutions were prepared by bubbling ultrapure argon gas (99.9999% purity) for 30 min. FMN was chemically reduced with a stoichiometric amount of sodium dithionite under argon atmosphere. Desired oxygen concentrations were achieved by mixing deoxygenated solutions with oxygen-saturated ones (1130 µM O_2_). In all cases, flavin reduction/oxidation was monitored at 450 nm. A flow speed of 7 mL·s^−1^ resulted in a dead time of 0.3 ms. Results from the stopped-flow experiments were analyzed in the Bio-Kine software (Bio-Logic, Claix, France). The values of *k*_obs_ were obtained using a monoexponential function from 6–8 repeats using Marquardt–Levenberg non-linear fit algorithms included in the Origin2021b software (OriginLab Corporation, Northampton, MA, USA). The *K*_NADH_, *k*_red_, *K*_O2_, and *k*_ox_ values were calculated from hyperbolic fit of the dependence of *k*_obs_ on concentrations of NADH and O_2_.

### 4.6. Stereospecificity Determination

*R*-NADD and *S*-NADD was prepared according to [6]. The specific activities were measured in 1 mL solution containing 0.1 M sodium phosphate buffer (pH 7.3), 0.15 mM stereospecifically deuterium-labelled NADH, 0.05 mM FMN, and 31 nM Pden_5119, 1.77 µM 5119-K82A, 5.24 µM 5119-K82H, or 4.37 µM 5119-H117A.

### 4.7. Dissociation Constants

Dissociation constants of the Pden_5119 protein and its mutant forms with FMN were determined from fluorescence titration with excitation at 280 nm by adding the coenzyme into 0.1 M sodium phosphate buffer (pH 7.3) with 632 nM 5119, 1.44 µM 5119-K82A, 753 nM 5119-R116A, or 681 nM 5119-H117A. Decreasing emission intensities at 340 nm were monitored on Luminescence Spectrophotometer LS-50B (Perkin-Elmer, Boston, MA, USA). The dissociation constants *K*_d_^FMN^ were calculated by non-linear regression analysis according to the equimolar binding model [34] using the Origin2021b software.

### 4.8. Crystallization, X-ray Data Collection, and Structure Refinement

The crystallization conditions that ensured the production of the best quality crystals are summarized in Appendix A. Synchrotron data were reduced using XDS [35] and scaled using SCALA-included CCP4 package [36]. The data are summarized in Appendix A. The refinement was performed in multiple cycles using REFMAC [37] as well as LORESTR [38], PDB REDO server [39], and PHENIX real space refine function [40]. The structure refinement is in Appendix A.

### 4.9. Small-Angle X-ray Scattering (SAXS)

The SAXS data collection are summarized in Appendix A. Merging and integral structural parameter determination was performed using PRIMUS/qt ATSAS 2.8.2 (r9678) [41]. Evaluation of the solution scattering of the atomic models and the fitting to experimental data was performed by CRYSOL v2.8.3 [42] using most probable dimeric assembly of homology model. Homology modeling was performed by SWISS-MODEL [43] using structure of FMN-reductase Msue from *Pseudomonas putida* (PDBID: 4C76) as a template. Dimeric structure of the homology model was assembled in the same way as two chains in the asymmetry unit of the crystal structure of the template (PDB ID: 4C76). Ab initio model was reconstructed using DAMMIN 5.3 [44].

### 4.10. In Silico Docking

Chain D of the ASU was selected for docking. The residues undefined in the electron density map and the necessary sidechains in the close vicinity of the binding site were added in using COOT [45]. Docking was performed using LeDock [46] with RMSD set to default 1 Å. After the initial docking, the results were evaluated and a score of possible rotamers and side minor sidechain rearrangements were tested until the final structure was obtained consistent with the binding site.

## 5. Conclusions

The two main aims of our study were to learn more details of the catalysis by Pden_5119 and to shed light on the molecular basis for the difference in biological activity of Pden_5119 and its structural homologues. Our results reveal the importance of three specific amino acid residues located at or near the flavin binding site. His-117 is implicated in binding the isoalloxazine ring of FMN and fixing it in a proper position. Lys-82 interacts with NADH and guarantees the correct stereochemistry of the hydride transfer from NADH to the bound FMN. The positive charge of Arg-116 is critical for the oxygen reactivity of the reduced flavin. To unravel the details of oxygen activation is an important goal for future studies.

## Figures and Tables

**Figure 1 ijms-24-03732-f001:**
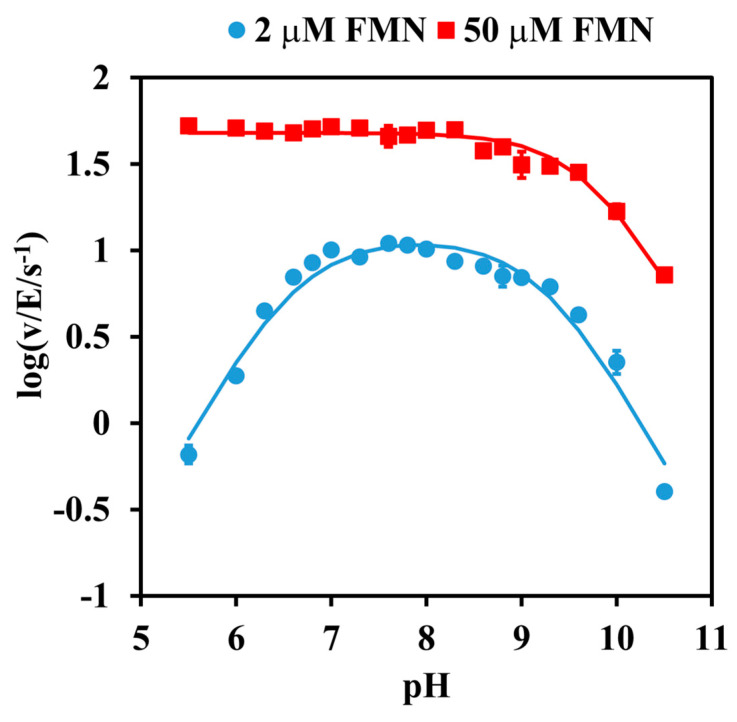
pH-rate profile for the Pden_5119-catalyzed oxidation of NADH. The initial rate measurement was started by addition of 0.38 μM enzyme to 40 mM Britton-Robinson buffer of indicated pH, 30 °C, containing 150 μM NADH and 2 μM (circles) or 50 μM (squares) FMN. Data points represent means of technical triplicates. Error bars show ± relative standard errors of v/E divided by ln10. They are omitted when they are comparable to or smaller than the symbol size. The lines are the theoretical curves corresponding to a single pKa of 9.72 (upper curve) and two pKas of 6.63 and 9.21 (lower curve).

**Figure 2 ijms-24-03732-f002:**
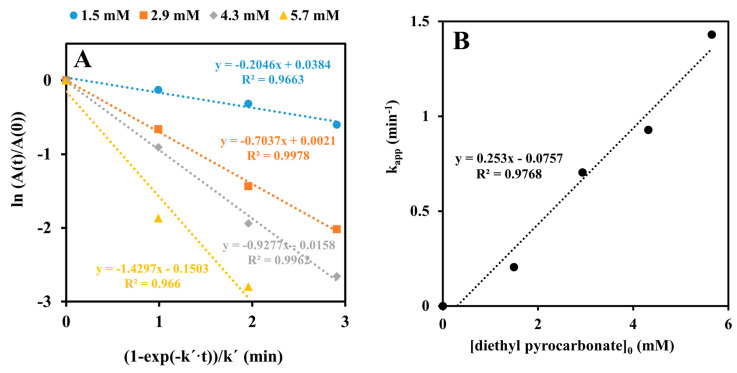
(**A**) Semi-logarithmic plot of time courses of inactivation of Pden_5119 at different concentrations of diethyl pyrocarbonate with correction for reagent decomposition. Reactions were performed on ice in 100 μL of 0.1 M sodium phosphate buffer, pH 7.3, with 0.22 mM enzyme. Millimolar diethyl pyrocarbonate concentrations were 1.5, 2.9, 4.3, and 5.7. Then, 15 μL aliquots were withdrawn at 1 min time intervals, mixed rapidly with 15 μL of 20 mM imidazole in phosphate buffer to destroy any unreacted diethyl pyrocarbonate, and assayed for residual enzymatic activity. No loss of activity was observed during incubation of the enzyme in the absence of diethyl pyrocarbonate. The first-order rate constant for decomposition of diethyl pyrocarbonate, k’, was determined separately and amounted to 0.0203 min^−1^. (**B**) Plot of the *k_app_* values, derived from the slopes of the lines in (**A**), versus diethyl pyrocarbonate concentration. The rate constant, k, for inactivation is equal to the slope of the regression line which is 0.25 mM^−1^·min^−1^ with a standard error of 0.02.

**Figure 3 ijms-24-03732-f003:**
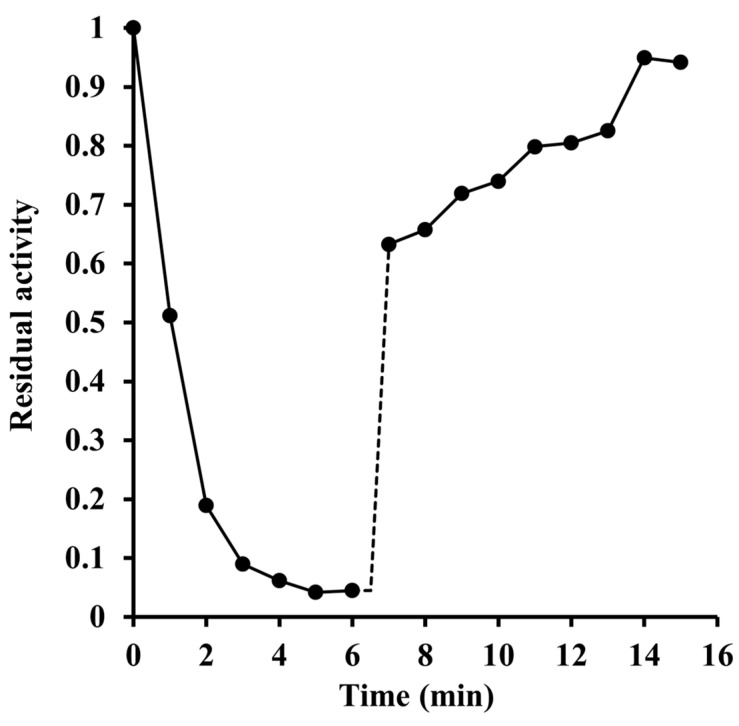
Inactivation of Pden_5119 by diethyl pyrocarbonate and reactivation by hydroxylamine. First, 0.27 mM enzyme was incubated with 2.6 mM diethyl pyrocarbonate in 205 μL of 0.1 M sodium phosphate buffer pH 7.3 at 0 °C. Then, 5 μL samples were taken at 1 min intervals to measure the residual activity which was expressed relative to the initial activity for non-incubated enzyme. At 6.5 min, 20 μL of 0.2 M hydroxylamine hydrochloride (adjusted to pH 7.3 with NaOH) was added to give a final concentration of 13.8 mM and the activity was further monitored until the fifteenth minute.

**Figure 4 ijms-24-03732-f004:**
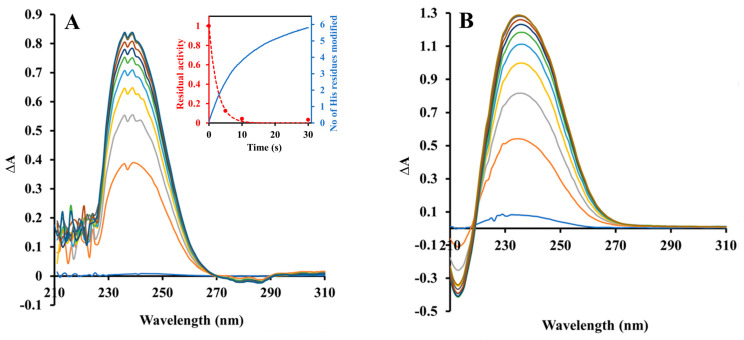
Ultraviolet difference absorption spectra of the Pden_5519 protein (**A**) and histidine (**B**) treated by diethyl pyrocarbonate. The modification reactions took place in a 1 cm quartz cuvette thermostated to 10 °C containing 22 μM protein or 400 μM histidine in 1 mL of 0.1 M sodium phosphate buffer, pH 7.3, and was initiated by adding 40 μL of a 0.26 M stock ethanolic solution of diethyl pyrocarbonate (final concentration 5.2 mM). Difference spectra were obtained by subtracting the spectrum before diethyl pyrocarbonate treatment. The scan time was 1 s and the delay between scans 30 s. The inset in panel A compares the time course of degree of enzyme modification with the loss of enzymatic activity, determined in a parallel experiment under identical conditions by taking of 20 μL samples for activity measurement.

**Figure 5 ijms-24-03732-f005:**
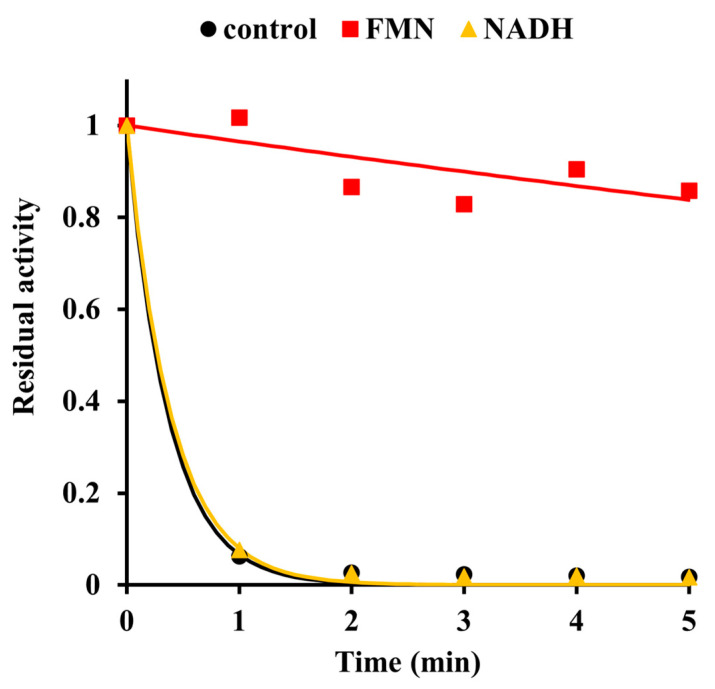
Substrate protection against inactivation by diethyl pyrocarbonate. First, 62 μL reaction mixtures containing 0.23 mM Pden_5119 with either no substrate or with 1.6 mM of either FMN or NADH in 0.1 M sodium phosphate, pH 7.3 at 0 °C were incubated with 2.2 mM diethyl pyrocarbonate. Aliquots (10 μL) were taken at 1 min intervals, quenched with an equal volume of phosphate buffer with 20 mM imidazole and assayed for activity. The activities are normalized relative to the initial activity.

**Figure 6 ijms-24-03732-f006:**
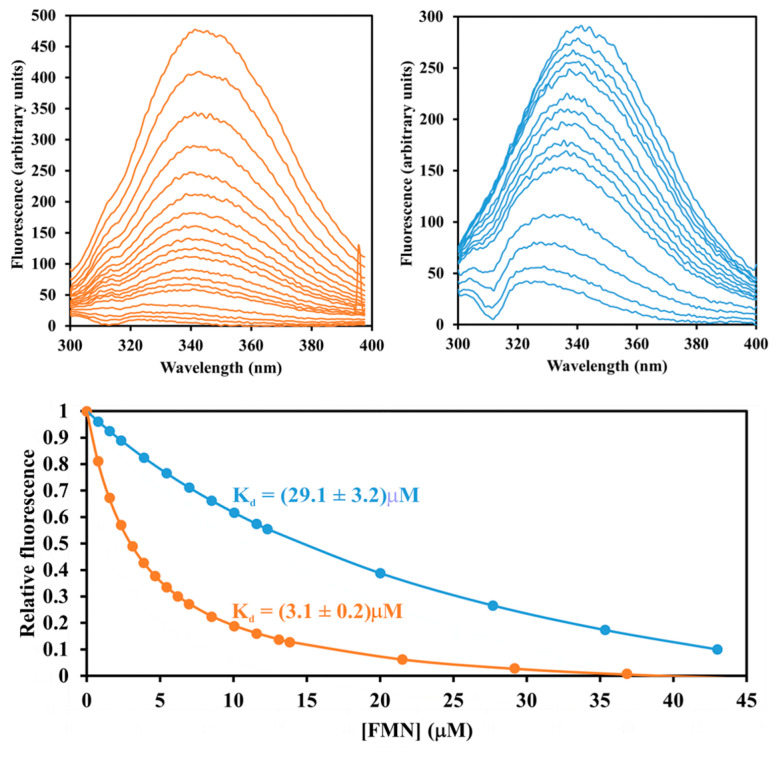
Fluorometric analysis of FMN binding to the native and diethyl pyrocarbonate modified Pden_5119 protein. After 5 min modification with or without 5 mM diethyl pyrocarbonate, 0.66 μM protein in 3 mL of 0.1 M sodium phosphate buffer (pH 7.3) was titrated by adding 2.5 μL portions of FMN (10 mM stock concentration) at room temperature. The FMN–protein interaction was monitored by recording fluorescence emission spectra upon excitation at 280 nm (upper panels, left—untreated protein, right—protein treated with diethyl pyrocarbonate). Changes in the relative fluorescence intensity at 340 nm were fit with a 1:1 binding isotherm. The best fits are represented by solid lines in the lower panel together with the corresponding values of the dissociation constants ± SD.

**Figure 7 ijms-24-03732-f007:**
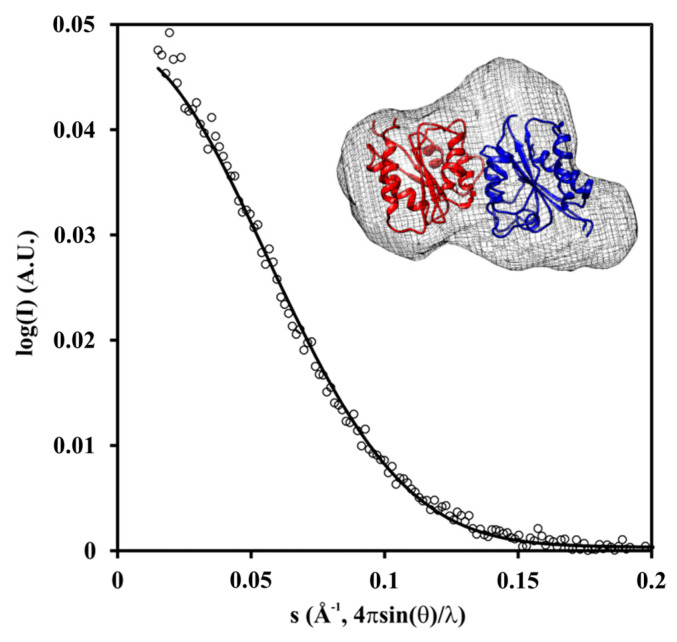
SAXS scattering curves. X-ray scattering of the 5119 protein (open circles) was compared with calculated scattering of the dimer (solid line) computed by CRYSOL with good overall fit (χ^2^ = 0.946) for the concentration 1.5 mg·mL^−1^.

**Figure 8 ijms-24-03732-f008:**
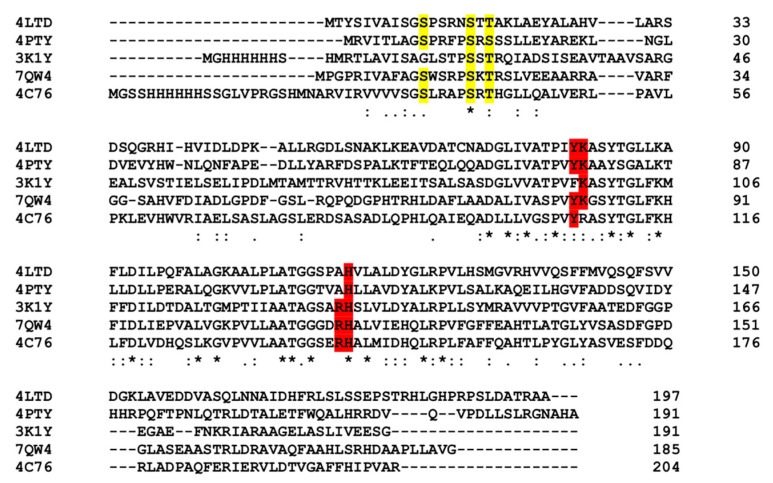
Clustal comparison of Pden_5119 (PDB ID: 7QW4) to closest-matching proteins. Asterisks denote identity, colons strong similarity, and periods weak similarity. Highlighted in yellow are residues implicated in binding the phosphate group of FMN [16]. Residues highlighted in red are those in the direct vicinity of the active site.

**Figure 9 ijms-24-03732-f009:**
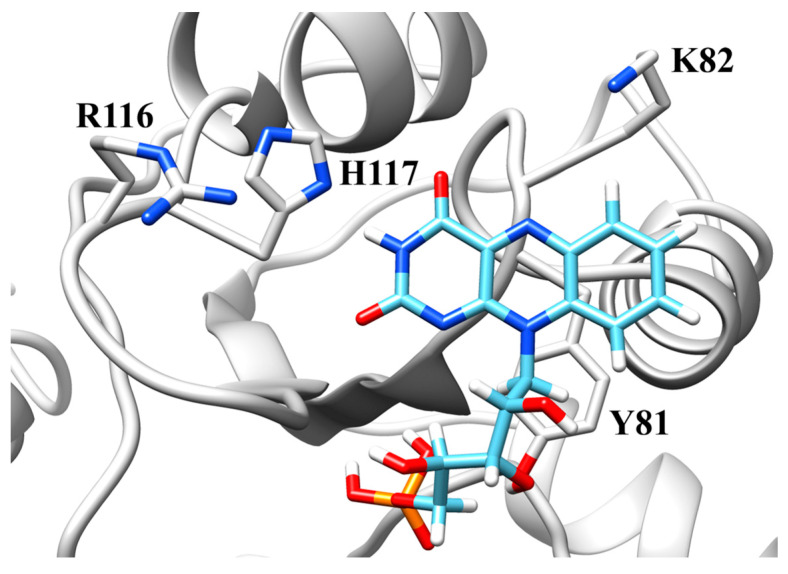
Putative active site of Pden_5119 showing the bound FMN and amino acid residues potentially important for catalysis. The figure was drawn with Chimera [17].

**Figure 10 ijms-24-03732-f010:**
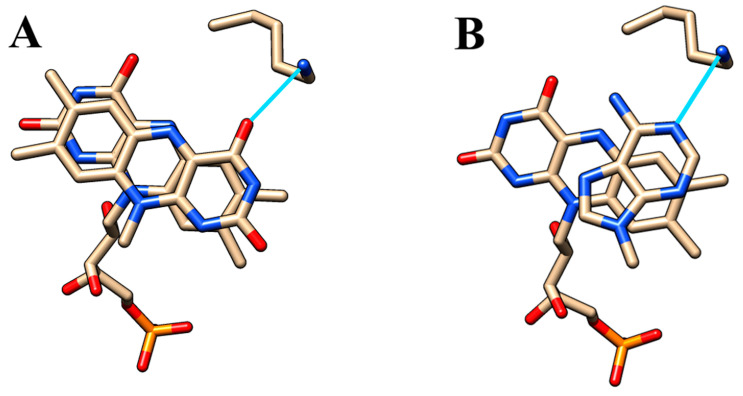
Participation of the conserved lysine residue in the binding of substrates to the flavin cofactor. Depicted are relevant structural parts of the complexes EmoB-FMN-FMN (**A**), EmoB-FMN-NADH (**B**), and SsuE-FMN-FMN (**C**). The line represents a hydrogen bond involving the amino group of the lysine residue (**D**), the postulated mode of binding of the nicotinamide ring of NADH. The figures were drawn with Chimera [17].

**Table 1 ijms-24-03732-t001:** Blast and DALI comparison of Pden_5119 (PDB ID: 7QW4) to closest-matching proteins.

PDB ID	Expect Value	Identities	Z-Score	rmsd
4C76	2 × 10^−42^	76/165 (46%)	23.1	1.5
3K1Y	6 × 10^−32^	62/171(36%)	21.9	2.1
4LTD	6 × 10^−26^	63/172 (37%)	21.3	2.1
4PTY	8 × 10^−21^	58/179 (32%)	21.2	1.9

**Table 2 ijms-24-03732-t002:** Thermodynamic and kinetic parameters of Pden_5119 and the site-directed mutants. The numbers are best-fit parameter values ± standard deviation as determined by non-linear regression.

	*K*_d_^FMN^ (µM)	*K*_NADH_ (µM)	*k*_red_ (s^−1^)	*K*_O2_ (µM)	*k*_ox_ (s^−1^)	*k*_cat_ (s^−1^)
Wild-type	3.8 ± 0.4	19 ± 1	41 ± 1	30 ± 3	24 ± 1	21 ± 1
K82A	6.8 ± 3.4	53 ± 18	0.78 ± 0.05	18 ± 5	5.8 ± 0.4	0.62 ± 0.05
R116A	3.5 ± 1.5	39 ± 17	6.5 ± 0.5	75 ± 2	0.91 ± 0.01	0.79 ± 0.05
H117A	9.1 ± 0.6	40 ± 7	2.5 ± 0.1	70 ± 16	0.79 ± 0.05	0.74 ± 0.06
R116K	0.6 ± 0.1	20 ± 4	41 ± 2	34 ± 2	26 ± 6	17 ± 2

**Table 3 ijms-24-03732-t003:** Specific activities of the Pden_5119 protein and its mutants in the presence of the selectively deuterated NADH substrates. The numbers are means ± standard deviation from three replicates.

	Wild-Type (µmol·s^−1^·mg^−1^)	K82A (µmol·s^−1^·mg^−1^)	R116A (µmol·s^−1^·mg^−1^)	H117A (µmol·s^−1^·mg^−1^)
NADH	1.039 ± 0.051	0.0303 ± 0.0024	0.0396 ± 0.0024	0.036 ± 0.003
*R*-NADD	1.012 ± 0.036	0.0309 ± 0.028	0.0372 ± 0.0035	0.034 ± 0.007
*S*-NADD	0.422 ± 0.010	0.0296 ± 0.0005	0.0144 ± 0.0010	0.009 ± 0.001

## Data Availability

The data are contained within the article or supplementary material.

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
