# Peer review of "Structural Insight into Catalysis by the Flavin-Dependent NADH Oxidase (Pden_5119) of Paracoccus denitrificans"

_ijms, 2023, doi:10.3390/ijms24043732_

Round 1
Reviewer 1 Report
Dear Authors
A very well prepared study. No remarks.
The supplemental file could have a short table of contents or intro text to provide a view what the reader can expect.
Reviewer 2 Report
The manuscript " Structural Insight into Catalysis by the Flavin-Dependent NADH Oxidase (Pden_5119) of Paracoccus denitrificans" by Kryl Sedláček, and Kučera describes some detailled insight into the named NADH oxidase and seems to be a follow-up investigation to previously published work (ref. 8 in the manuscript: Sedláček, V. and Kučera, I. (2019), Functional and mechanistic characterization of an atypical flavin reductase encoded by the pden_5119 gene in Paracoccus denitrificans. Mol Microbiol, 112: 166-183). In this manuscript, the focus is laid on distinct amino acids that are crucial for the enzymatic reaction and the effects that targeted inhibiton or exchange of potentially crucial amino acids excert on the overall enzymatic activity. The manuscript is, as a whole, well written and comprehensible, and certainly interesting to the specialized reader. I recommend publication of the manuscript after a some optimizations. Some data is missing and should be added, as listed below.
General / scientific points:
Figures: Albeit some info on errors is given in most figure captions, standard deviation and error bars are generally missing in all figures. As typically, these kinds of experiments (Figs 1-6) are done in biological replicates, the data should be supplemented accordingly and the respective nformation (on replicates etc) included in the methods section.
Activity assays: How was the enzyme concentration determined and decided how much enzyme is used in each assay? It is striking that enzyme concentration (Which is not named in the methods section and should be!) varies with each experiment whilst the other conditions are rather constant. One exception is figure 5, where even different buffer concentrations are used. Why? Furthermore, no R2 values or regression formulae are given for the data. Please add - either as table in the supporting file or within the figures.
L85: Why is the data not shown? If it is important for the understanding of the research it should be added - at least in the supporting file.
L94/95: same here - observations are claimed that have no data to underpin it. Please add.
Figure 3: Even though the experiment is more of a qualitative work, the concentration changes that happen to the enzyme during addition of 20uL NH2OH to 175uL residual volume should be taken into consideration. As there is no mentioning of it I suppose this was not the case. Please correct the data. Was the pH checked and the buffer capacity high enough?
Figs 2 and 5: auto-degradation constants of the inhibitors are stated but their determination not described. Please add.
Fig 5 and 6: if 6 resp. 7 x 5uL aliquots are withdrawn from 40uL experiments, "constant" reaction conditions do not seem to be maintained. Effects of mixing, shearing, temperature distribution or O2 supply - and others - might be affected. Please explain the reasoning behind the experimental design.
Fig. 8: detailled information on the tree parameters are not given. As such, the tree contains no information on homology nor is it novel - with respect to the sequence comparisons reported in the authors' ealrier work. In this context, I advise to move it to the supporting information and give more details there. The main manuscript wull work easily with only the alignment from table 1 /Fig9.
Section 2.5: This docking data clearly puts the reasoning behind the experiments reported in sections 2.1 and 2.2. I wonder if the manuscripts' story would be easier to follow if the X-ray structure and amino acid sequence would be reported first, followed by all the functional experiments? Bsically putting the order 2.3-2.4-2.5-2.1-2.2-2.6....? It might be something to consider.
Section 2.6, L220ff: Here, CD spectroscopy is mentioned, but no data or further information is given. why? If it is a crucial point to the scientifc story, please add the data or give a reference. Otherwise remove the claim.
Section 2.8: In L266, kinetic data is given. Why is it not added to table 2? That would make it far better comprehendable. Also, inactivation effects are claimed (L267) but no data is given. Please add.
Methods section: Certainly, a previously described enzyme production method should be referenced and not repeatedly reported. However, information on the mutant generation and their production/purification cannot be found in the cited references and should be described properly.
Minor / formal points:
L46: insert commma: "...both, flavin reductases,"
L48: improve wording in "FMN adds before NADH", e.g. to FMN binds before NADH (or equivalent)
Tables 2/3: the labelling of the mutants is inconsitent. Please use only one version.
L294: add "act" in "and () can as a donor"
Fig 11: This figure should be reduced in size.
L384/5: optimize wording in "made up by..." --> made, prepared...
Sections 4.3, 4.4, 4.5: Please be more detailled in the measurement parameters. Better list a recording wavelenth once too often than not enough.
L415 condition should be plural.
Reviewer 3 Report
This article describes the characterization of a bacterial redox enzyme with functional features of both flavin reductases and flavin oxidases.
This work includes the identification of residues important for cofactor binding or catalysis by chemical modification, and the determination of the crystal structure of the enzyme. Next, as co-crystalls of the enzyme and FMN coud not be obtained the authors used an in silico docking analysis to model the position the cofactor in the active site. This was followed by a site-directed mutagenesis study of putative active site residues and the characterization of the FMN binding properties and enzymatic activity of the mutants, including stopped-flow kinetics. This latter analysis enabled the authors to determine the function of these residues in the various parts of the flavin cycle, which was confirmed by using deuterated derivatives of NADH.
This is a well thought-out piece of work, performed by specialists of this type of enzymes. The discussion is also appropriate. I have no major criticisms on this work.
Author Response
We thank the reviewer for the revision work.
Round 2
Reviewer 2 Report
Thank you for adressing my earlier raised comments. I am fine with the revides manuscript.
Author Response
We thank the reviewer for his/her careful reading of the manuscript and the constructive remarks.
Reviewer 4 Report
line 196: 3K1Y
line 221: His-117
line 238 (Table 2): Kd FMN
line 237 (Fig. S7): The intensities of the CD signals of wild-type enzyme and active site variants vary significantly. Does this mean that the active site variants are partially unfolded (up to 50% in R116A as judged from the signal between 210 and 220 nm) or is there something wrong with the protein concentrations?
